# Predicting DNA structure using a deep learning method

Jinsen Li [1], Tsu-Pei Chiu [1] & Remo Rohs [1,2,3,4] ✉

Understanding the mechanisms of protein-DNA binding is critical in comprehending gene regulation. Three-dimensional DNA structure, also described as DNA shape, plays a key role in these mechanisms. In this study, we present a deep learning-based method, Deep DNAshape, that fundamentally changes the current *k*-mer based high-throughput prediction of DNA shape features by accurately accounting for the influence of extended flanking regions, without the need for extensive molecular simulations or structural biology experiments. By using the Deep DNAshape method, DNA structural features can be predicted for any length and number of DNA sequences in a high-throughput manner, providing an understanding of the effects of flanking regions on DNA structure in a target region of a sequence. The Deep DNAshape method provides access to the influence of distant flanking regions on a region of interest. Our findings reveal that DNA shape readout mechanisms of a core target are quantitatively affected by flanking regions, including extended flanking regions, providing valuable insights into the detailed structural readout mechanisms of protein-DNA binding. Furthermore, when incorporated in machine learning models, the features generated by Deep DNAshape improve the model prediction accuracy. Collectively, Deep DNAshape can serve as versatile and powerful tool for diverse DNA structure-related studies.

Binding interactions between DNA binding proteins such as transcription factors (TFs) and their DNA target sites are crucial for gene regulation; thus, it is vital to fully understand TF-DNA binding mechanisms. These mechanisms can be viewed from two perspectives: base readout and shape readout[1,2]. Base readout occurs through direct contacts between amino acids and DNA bases, and a relatively strict pattern is followed. For instance, the zinc finger TF family heavily relies on base readout to recognize DNA targets[3,4]. Shape readout happens when proteins interact with the double helical conformation, rather than interacting directly through hydrogen bonds with certain atoms on nucleobases. As an example, proteins containing positively charged amino acids (e.g., arginine, protonated histidine, and lysine) may favor a certain DNA structural feature, such as a narrower minor groove that is more negatively charged[1]. Some TFs from the

homeodomain family utilize shape readout in specific DNA regions within the core binding site[5–9]. Narrower minor groove regions can result from diverse DNA sequences, providing an additional signal for binding specificity. Another example is the myocyte enhancer factor-2 (MEF2) family, which uses shape readout in the center of its degenerate motif[10]. In general, TFs use a combination of base and shape readout modes to achieve DNA binding specificity. The mismatch-induced DNA structural alterations that influence TF-DNA binding affinity prove the importance of shape readout[11]. However, the influence of the base and shape readout mechanisms can be difficult to parse.

The influence of shape readout in DNA binding mechanisms extends to various aspects of DNA structural features, also known as DNA shape features. To measure shape readout quantitatively, one must first define DNA shape. This article focuses on the B-form

[1]Department of Quantitative and Computational Biology, University of Southern California, Los Angeles, CA 90089, USA. [2]Department of Chemistry, University of Southern California, Los Angeles, CA 90089, USA. [3]Department of Physics and Astronomy, University of Southern California, Los Angeles, CA 90089, USA. [4]Thomas Lord Department of Computer Science, University of Southern California, Los Angeles, CA 90089, USA. ✉e-mail: rohs@usc.edu

representation of DNA molecules - the canonical right-handed double helix. Three-dimensional (3D) DNA shape features describe the global and local relationships between nucleobases, base pairs, and sugar-phosphate backbones[12]. Although DNA oligonucleotides are flexible molecules with varying DNA shape features, certain optimal conformations may be intrinsically favorable, as seen in their static or sometimes equivalent time-averaged representation of DNA shape features[13]. Proteins that favor certain intrinsic DNA shape features may bind DNA targets with other DNA structures; however, they may require a higher energy to maintain the binding, thus exhibiting a lower binding affinity[6].

The energy required to alter an unfavored static DNA structure may depend on the DNA flexibility. Therefore, it is important to consider the conformational flexibility of the DNA and fluctuations in its shape features. As revealed by, for instance, molecular dynamics (MD) simulations, DNA shape features fluctuate in various ways[14–16]. For example, CpG, CpA and TpA base-pair (bp) steps are generally more flexible[17], whereas A-tracts consisting of consecutive ApA, ApT, or TpT bp steps are conformationally rigid[17,18]. The length of A-tracts has also been observed to impact the flexibility of neighboring regions[19]. Methylated cytosine induces DNA flexibility changes that in turn influence nucleosome stability[20,21]. Another recent study using MD simulations showed that conformational flexibility contributed by the flanks plays an important role in homeodomain binding[22,23]. However, no high-throughput methods have been proposed yet to predict DNA shape fluctuations.

Methods such as MD or Monte-Carlo (MC) simulations and X-ray crystallography (XRC) can be used to acquire DNA structures for short DNA fragments. Methods such as Curves[12,24,25] and 3DNA[26] have been developed to derive DNA shape features from computational trajectories or experimentally solved structures. High-throughput methods such as DNAshape[27–29] have been introduced to circumvent the difficult and sometimes impossible task of simulating or experimentally solving structures. Numerous studies have successfully employed the DNAshape method[27], demonstrating the effectiveness of using high-throughput methods to predict DNA shape features[30–34].

Nevertheless, although DNAshape[27–29] is an efficient method, it relies entirely on a pentamer query table containing all possible pentamers compiled from extensive MC simulations[35]. The pentamer length limits this method because only the nearest and next-nearest neighbors are accounted for when considering the influence of the sequence environment on the center of the pentamer. As for other data sources, such as MD simulation data and experimental structures in the Protein Data Bank (PDB)[36], only tetramer query tables could be generated due to data paucity[28]. Therefore, effects from longer-range neighbors are totally neglected in this query table setup.

Our approach, Deep DNAshape, overcomes the limitation of DNAshape, particularly its reliance on the query table search key. This advancement is pivotal, given that the limitation was only caused by the available amount of data. Deep DNAshape enhances the capability to discern how the shape at the center of a pentamer region is influenced by its extended flanking regions, providing a model that offers a more accurate representation of DNA.

The pentamer query table contains all possible pentamers for DNA shape features of the central bp, considering up to a 2-bp flanking region, which consists of the nearest-neighbors and next-nearest-neighbors. Despite this definition, flanking regions exceeding two bp may still be influential, which has been shown for the central TpA step[37]. For some TFs having a long core motif, a more complete view of DNA shape considering longer-range flanking regions is necessary. In addition, some DNA shape parameters are bimodal[15], and statistically calculating an average value for the query table may not capture the whole picture. The ability to approximate DNA shape features using only sequence information such as mono- and dinucleotides[38] also

highlights the limitations of the pentamer query table. Although it is possible to generate a query table for longer $k$-mers, the number of simulations that would be necessary to cover all longer $k$-mers is exponentially higher; meanwhile, there are not enough existing experimental structures[28]. Therefore, there is a need for a method that can accurately predict DNA shape features in a high-throughput manner using only limited data while considering longer-range effects.

To develop such a model, we began with assumptions about how the 3D DNA structure is affected by sequence. Firstly, we assumed that DNA shape features at each bp are mainly influenced by their neighboring bp, and that this influence weakens with distance. Secondly, we assumed that this influence can be statistically inferred. Therefore, we designed a specialized deep learning architecture to deal with variable-length DNA sequences and computed the neighboring effects of flanking regions in a layer-by-layer manner (Fig. 1a–d, Supplementary Fig. 1). We trained the model on DNA shape features that were previously analyzed and compiled from MC simulations (Fig. 1b, Supplementary Figs. 2–3) and which had been experimentally validated[27]. The model now considers longer range neighboring effects compared to current data source limitations (Fig. 1e) and can be used for predicting DNA shape features from any given sequence (Fig. 1f, g). We then evaluated the predicted DNA shape features with a tetramer query table derived from MD simulations (Supplementary Table 1)[28]. We compared predictions from our resulting model, Deep DNAshape, to predictions from our original pentamer-based DNAshape (DNAshapeR)[39] method (Figs. 2–4, Supplementary Table 2). To thoroughly benchmark the model, we also trained it on alternative DNA shape sourced from experimentally solved structures[40] and MD simulations[41] (see Supplementary Information), leading to the generation of Deep DNAshape (Expt) and Deep DNAshape (MD). These models were then compared with the MC-trained Deep DNAshape method, along with comparative analysis across all model variants (Supplementary Fig. 4, Supplementary Table 3).

In addition, the design of the model unlocks the potential to examine systematically how DNA shape fluctuations are affected by extended flanking regions. We previously generated a query table containing standard deviation (SD) values for 13 DNA shape features and used them in a machine learning study[28]. Although these values were statistically computed, nevertheless, the model performance was significantly improved when the values were included[28]. We assumed that these SD values were highly correlated with true fluctuation values. Therefore, although conformational flexibility of DNA is important[42], it is the fluctuation of shape features that is difficult to access and frequently overlooked in research. Here, we used the same approach as in predicting the static DNA shape to predict DNA shape fluctuation (FL) values: specifically, we directly calculated fluctuation from MC simulations using Curves[12]. We investigated whether these high-throughput-predicted fluctuation values aligned with previous findings on DNA flexibility[42], and we considered the insights that these fluctuation values might provide.

Next, we compared the Deep DNAshape model with the original pentamer-based DNAshape method. We tested our model on data from TF-DNA binding assays for quantifying the relative binding affinity of DNA sequences for any given TF. These binding assays consisted of multiple in vitro experimental methods, such as protein binding microarray (PBM)[43], HT-SELEX[44] and SELEX-seq[7]. We previously used these datasets in conjunction with an expanded set of 13 DNA shape features including groove features, inter-bp features, and intra-bp features (see Methods), demonstrating the effectiveness of DNA shape features in predicting TF-DNA binding specificities using L2-regularized multiple linear regression[28]. We tested DNA shape predicted by Deep DNAshape against the same datasets to determine if improvements could be detected compared to the original pentamer DNAshape model (Fig. 5). Finally, we showed the potential of Deep DNAshape by processing large genomic-level data.

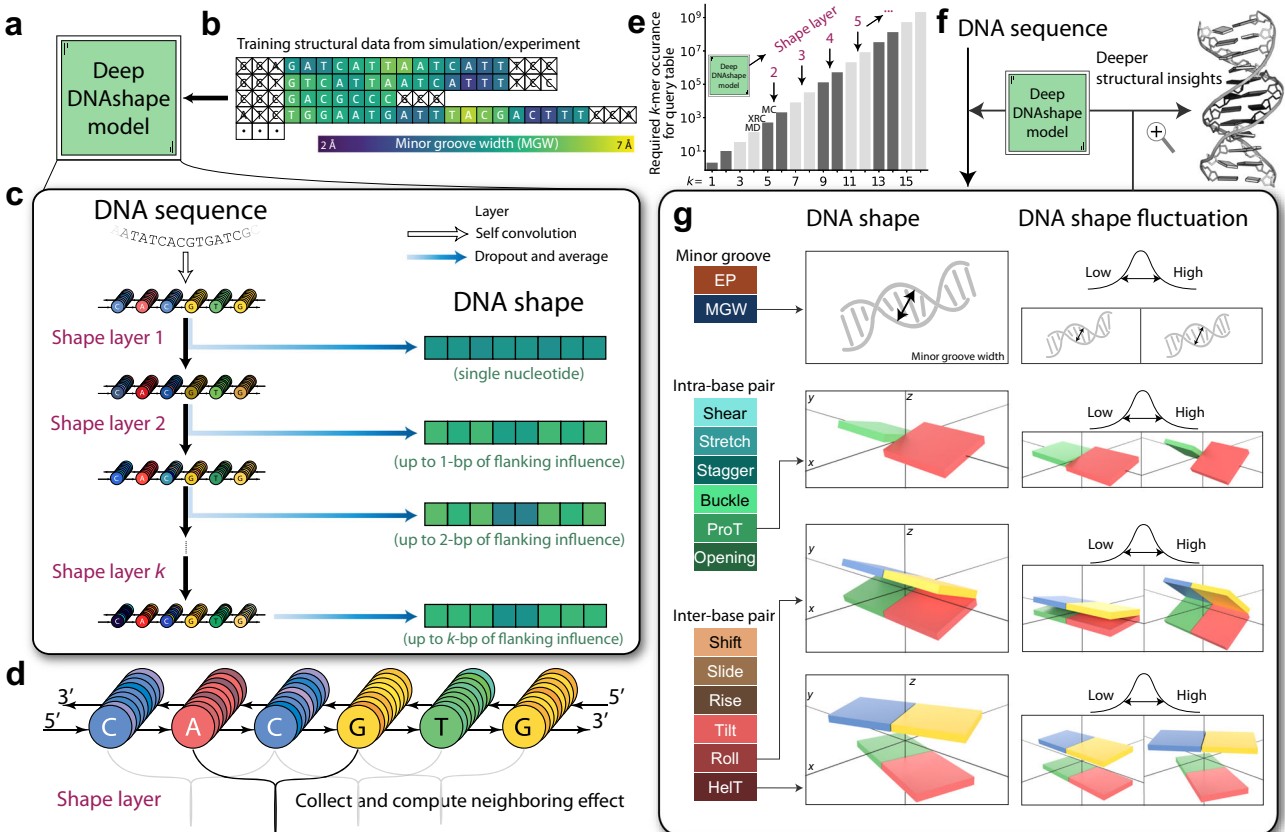

**Fig. 1 | Deep DNAshape schematic.** Deep DNAshape model (**a**) and training data for the model (**b**). Sequences used in simulations are pooled together, and corresponding DNA shape values are assigned to each position of each sequence. Some values, such as MGW at the terminal bp, are not defined and therefore excluded. **c** Schematic representation of the Deep DNAshape model. DNA sequences go through several 'shape layers'. Each time the model goes through a shape layer, features for each individual position are passed to its nearby two positions, allowing one additional consideration for the flanking region. **d** Simple diagram of 'shape layer'. In each layer, features from neighboring nodes are collected and computed with information in each current node. Features are updated for each node on the sequence. See Methods for details. **e** Capacity comparison between the

Deep DNAshape model and current data limitations. The shown labels 'MC', 'XRC' and 'MD' are current limitations to generate pentamer or tetramer query tables sourced from Monte Carlo simulations, X-ray crystallography, and molecular dynamics simulations, respectively. **f** Deep DNAshape model usage. Deep DNA-shape can process a given DNA sequence as a string of characters (A, C, G, and T) and predict any specific DNA shape feature for each nucleotide position of a sequence. **g** All DNA shape predicted by the Deep DNAshape model. In addition to static DNA shape features, the model can predict DNA shape fluctuations (Cartoons do not represent real values of low or high. Values can be negative or positive for angular shape parameters.). Shown are graphical explanations of the four most frequently-used DNA shape features (MGW, ProT, Roll, and HelT).

## Results

### Deep DNAshape predicts DNA shape and shape fluctuations considering extended flanking influences without biases

MC simulations were used to generate 3D structural predictions for numerous DNA samples with analyzed DNA shape values for 2121 different sequences[27]. For each individual DNA shape feature, a training file containing all sequences and their sequence–position-wise DNA shape values was compiled from the simulation data. Despite the varying lengths of these sequences, the Deep DNAshape model was designed to accommodate such variation (Fig. 1a–d, Supplementary Fig. 1). Following hyperparameter searches on training and validation data (Supplementary Figs. 2, 3), each model was trained on the entire training data and, when used together, was able to predict any DNA shape feature for any length of DNA sequence. The predictions consider only the local bp information of nearby bp (Supplementary Fig. 1). The models can predict DNA shape features by considering up to 7-bp flanking regions, and the number of bp of flanking regions considered can be selected for optimal accuracy (Fig. 1e–g). Models were trained with minimal overfitting, while maintaining high accuracy in deeper layers, as evident from the training and validation split samples in the hyperparameter searches (Supplementary Figs. 2, 3).

We validated our predictions against tetramers compiled from MD simulations[15] (Supplementary Table 1) and our previous pentamer query table (Supplementary Table 2). Additionally, we validated our predictions by calculating and comparing the average inter-bp features for 10 dinucleotides and intra-bp features for A-T and C-G bp, considering all possible flanking regions predicted by the Deep DNA-shape model and its variants using different data sources (Supplementary Fig. 4, Supplementary Table 3; also refer to Supplementary Text). This approach eliminated the use of a query table and enabled us to predict DNA shape features affected by longer flanking regions, compared to using a forcibly generated hexamer or heptamer query table with large numbers of missing values (Fig. 2a–d, Supplementary Figs. 5–7, Supplementary Fig. 8 for Deep DNAshape (MD)). Using our Deep DNAshape method, the inferred DNA shape values based on effects of extended flanking regions (Fig. 2a–d, Supplementary Figs. 5–7) are almost perfectly aligned (Supplementary Table 2) with statistically computed values. Compared to an interpolated query table, our method corrects the biases from different distributions of $k$-mers and artifacts from molecular simulations or experiments using DNA fragments (Supplementary Fig. 9).

By performing high-throughput prediction of DNA shape features considering extended flanking regions, Deep DNAshape can be used to

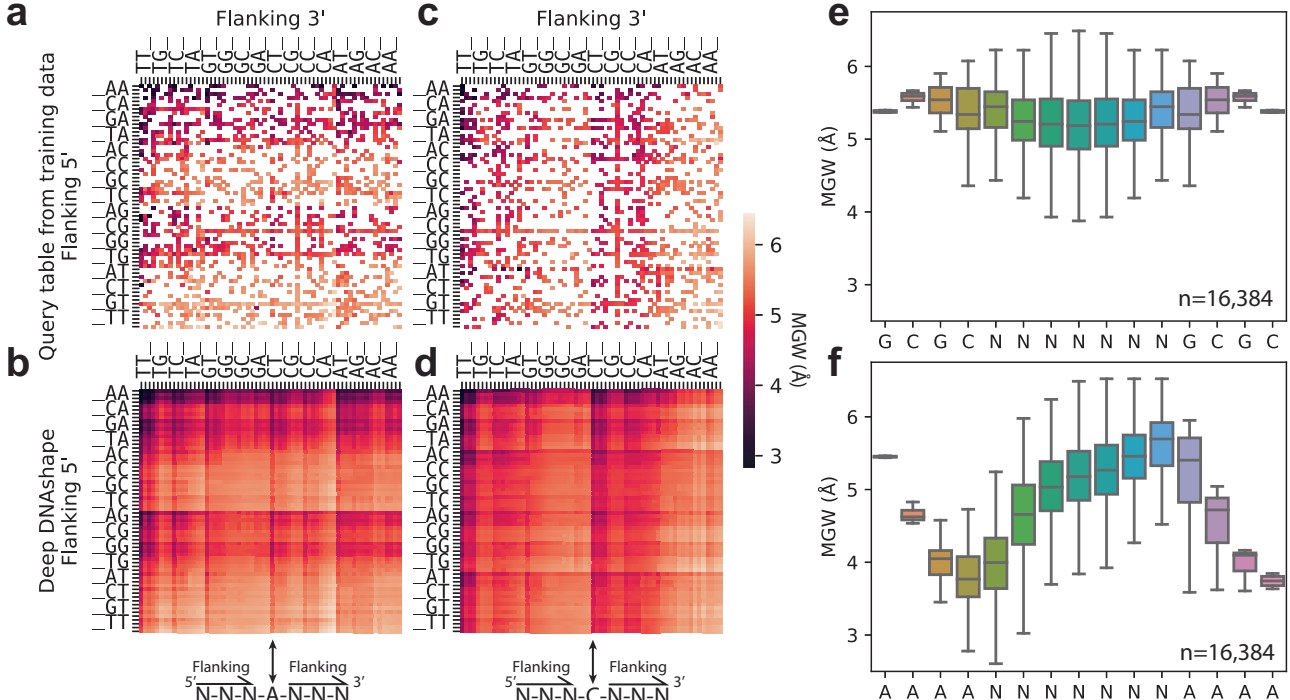

**Fig. 2 | Minor groove width predicted by Deep DNAshape using extended flanking regions. a**–**d** Heatmaps showing MGW values at central position for all possible 7-mers. **a**, **c** MGW values as if we constructed a 7-mer query table from all available MC simulations directly. **b**, **d** MGW predictions generated by the Deep DNAshape method for all possible 7-mers. **a**, **b** All sequences with 'A' in the center. **c**, **d** All sequences with 'C' in the center. Flanking regions from 5' and 3' are sorted based on distance to the center and bp character in the heatmap. '_' represents A, C, G, and T in sequential order for flanking 5' and in reverse order for flanking 3'. **e**–**f** Boxplots showing MGW values predicted by Deep DNAshape for random sequences with fixed 5' and 3' caps. **e** Predictions are capped by 'GCGC', and **f** Predictions are capped by 'AAAA'. Center line indicates the median. Box limits are 75th and 25th percentiles. The whiskers extend 1.5 times the interquartile range (IQR) from the top and bottom of the box. Outliers are removed in boxplots.

investigate the effects of flanking regions on the core DNA structure without requiring MD simulations or X-ray crystallography (XRC) experiments. We initially examined four DNA shape features, minor groove width (MGW), propeller twist (ProT), Roll, and helix twist (HelT), at the core of a DNA fragment for all sequence combinations, with the goal of examining the general neighboring effect on different cores (Supplementary Fig. 10). The static shape of the core can show bimodal or even trimodal distributions, affected by different flanking sequences (Supplementary Fig. 10).

One DNA fragment worth studying is the A-tract, which contains at least three consecutive ApA, ApT, or TpT bp steps without a flexible TpA bp step, and has been associated with a narrower MGW[1,9]. We permutated all *k*-mers with certain sequences as their cap, such as AAAA-NNNNNNN-AAAA, for 15-mers with A-tract caps. Next, we predicted the MGW using Deep DNAshape compared to counterparts capped by GCGC ends. This approach permitted us to see effects from different flanking cap combinations (Fig. 2e, f). The result indicated that A-tracts increase the flanking MGW on their 5' end and decrease the MGW on their 3' end. In other words, the MGW for a short strand of DNA will be upregulated at the 3' end and downregulated at the 5' end, if accompanied by two A-tracts at the 5' and 3' ends. The A-tracts showed a general trend of decreasing the MGW from 5' to 3', consistent with previous individual structural studies[45,46].

The dynamics of DNA sequences can be difficult to describe. In Deep DNAshape, DNA shape fluctuations can be predicted to represent conformational flexibility for an individual DNA molecule. The training data are based on a compiled shape fluctuation dataset from MC simulations. Different bp or bp steps have different intrinsic flexibilities. These fluctuation values are comparable to values that are directly computed from MD simulations (Supplementary Table 1)[15]. The values contribute to the global bendability, twistability, and

prolongability of DNA double helical molecules. The effects of A-tracts on fluctuations can also be visualized (Supplementary Fig. 11). We observed that flanking A-tracts greatly elevate the MGW fluctuations of the cores (Supplementary Fig. 11), whereas GCGC ends slightly decrease the MGW fluctuations of the cores.

## Deep DNAshape is superior to pentamer DNAshape in elucidating TF-DNA binding

High-throughput prediction of DNA shape can be applied to data from experimental binding assays on TF-DNA binding (Fig. 3a). The pentamer-based DNAshape method[27–29,39,47] lacked the ability to utilize longer-range flanking regions of sequences. Therefore, this method cannot diversify DNA shape changes in the core binding site affected by extended flanking regions (Fig. 3b) that may still contribute to TF binding specificity. For example, the DNA binding affinity of basic-helix-loop-helix (bHLH) TFs binding to enhancer boxes (E-boxes) (Fig. 3c) is greatly affected by the flanking regions[48]. The pentamer-based DNAshape method cannot access a flanking region-associated change in the DNA shape of the central 'CG' dinucleotide in the most common E-box motif 'CACGTG'. In the Deep DNAshape model, a 'shape layer' capable of utilizing extended flanking regions (Fig. 3b) may provide additional insights, such as into how bHLH proteins bind to their target DNA sequences.

We investigated genomic-context PBM data for three human bHLH TFs, Max, c-Myc, and Mad2. We plotted MGW predictions for the top 25% of aligned binding data (Fig. 3d, Supplementary Figs. 12, 13) and compared them to the predictions by the pentamer-based DNAshape method. Subtler differences can be seen with Deep DNAshape in the central motif cores than was possible with the pentamer-based DNAshape method, even though most of the cores had the same sequence identity 'CACGTG' (Fig. 3e–h). When we filtered out the TF-

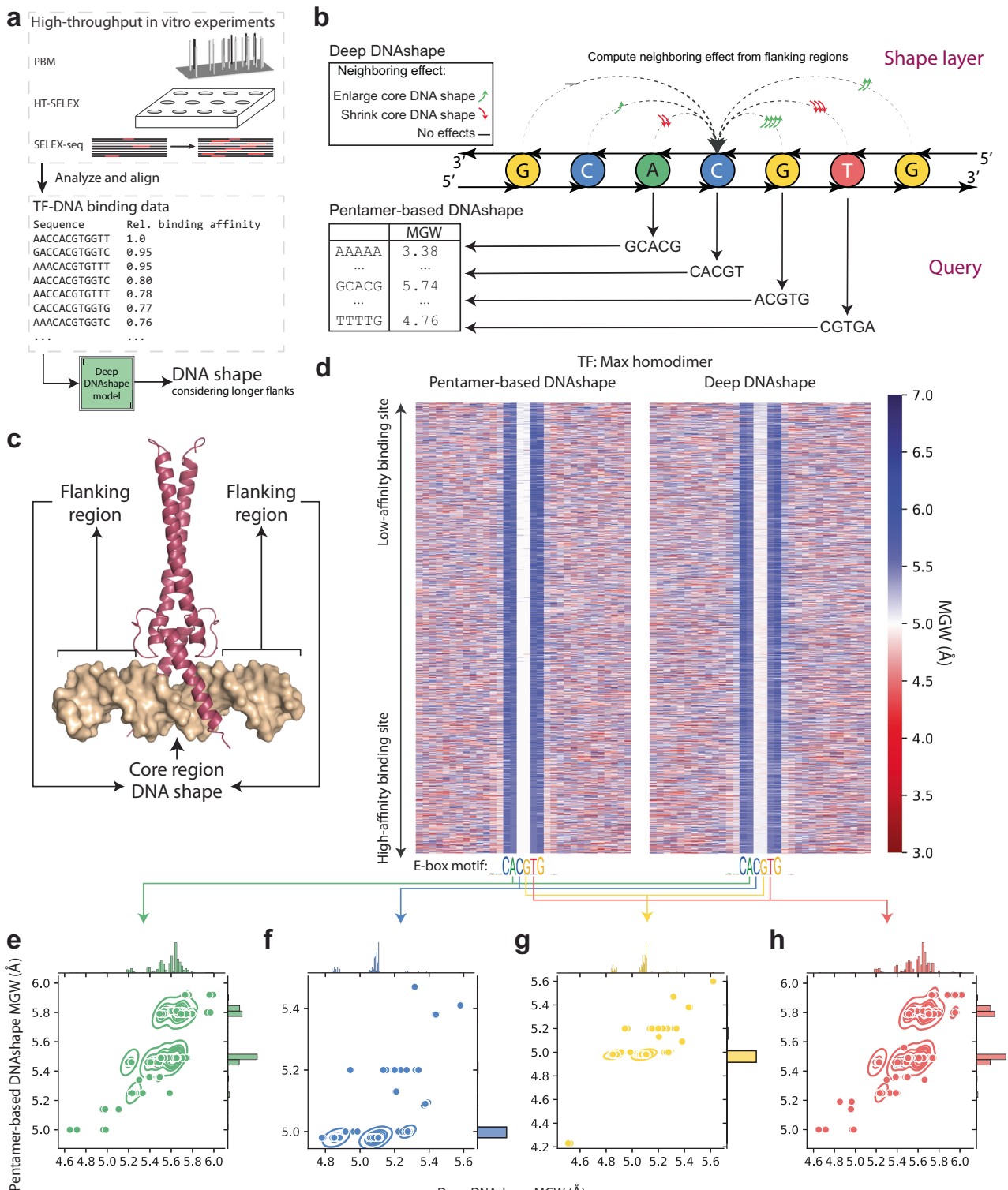

**Fig. 3 | Comparison of Deep DNAshape and pentamer-based DNAshape methods on TF-DNA binding data. a** General pipeline to apply Deep DNAshape on TF-DNA binding assay data. **b** Comparison of methodology to predict DNA shape features from Deep DNAshape and pentamer-based DNAshape methods. **c** Co-crystal structure of DNA-bound Max protein homodimer (PDB ID: 1AN2). Flanking regions indicate regions without protein contacts. Core is the region with protein contacts. **d** MGW predicted by pentamer-based DNAshape method vs. Deep DNAshape method for Max protein and DNA binding data, in order of relative binding affinity. Color represents DNA shape values. Data are aligned based on core binding site. Only top 25% of binding data are used. **e–h** Scatterplots of central four core MGW values predicted by Deep DNAshape and pentamer-based DNAshape. Gaussian kernel density estimation plot is added to show the contour of scatter density. Histograms along axes are shown to visualize the distribution of DNA shape values predicted by both methods. Values predicted by Deep DNAshape reveal structural variations at the center of the E-box that the pentamer-based DNAshape method was unable to reveal.

DNA binding dataset to include only sequences with 'CACGTG' in the core, we observed a consistent negative correlation between the binding affinities and the Roll and Roll-FL values predicted by Deep DNAshape (Supplementary Fig. 14), regardless of the in vitro experiment platform. These correlations are also observable using Deep DNAshape (Expt) (Supplementary Fig. 15a–c), but not when using Deep DNAshape (MD) (Supplementary Fig. 15d–f). Such visualizations had been unachievable with the pentamer-based DNAshape method because this method only considers 2-bp flanking regions (which, in the present case, remain constant). These results highlight the effect of the flanks and the need for a method that accounts for more than the

nearest- and next-nearest neighbors. Through Deep DNAshape, we can propose a hypothesis for the potential binding mechanism used by the bHLH family of TFs to distinguish identical binding cores (Supplementary Fig. 16), although this hypothesis will require further investigation to confirm.

We also investigated some of the well-studied homeodomain TFs from the homeobox (Hox) family (Fig. 4). Extradenticle (Exd) and Sex combs reduced (Scr) heterodimerize to bind to DNA sequences. The resulting Exd-Scr heterodimer (Fig. 4a–c) prefers a narrower MGW at the two 'AY' steps (Y: pyrimidine) in its motif 'NGAYNNAY'[7]. Another heterodimer, Exd with Ultrabithorax (Ubx),

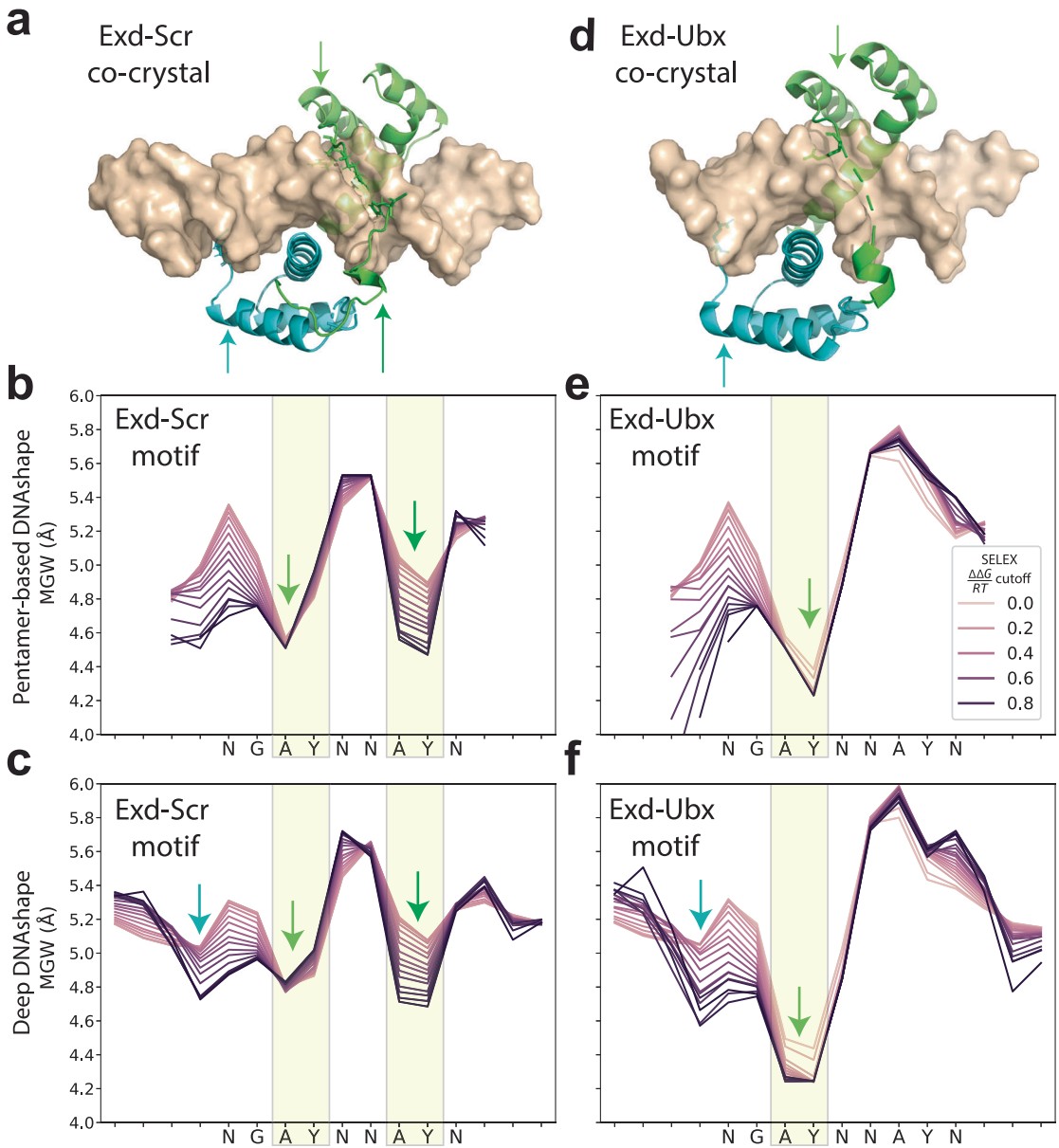

**Fig. 4 | Evaluations of Deep DNAshape on extended flanking regions using Hox-TF binding data. a** Co-crystal structure of DNA-bound Exd-Scr heterodimer (PDB ID: 2R5Z). Arrows indicate locations of insertion from Exd (cyan) and Scr (green) arginine residues into the minor groove. **b**, **c** MGW predicted by the pentamer-based DNAshape and Deep DNAshape methods, for Exd-Scr SELEX-seq data. Lines are calculated based on cutoff values of relative binding affinities. Darker colors in plot correspond to sequences with higher binding affinities. The comparison of the two panels shows that Deep DNAshape predicts the MGW of the flanking regions and the additional Exd minimum in MGW that the pentamer-based DNAshape

method was unable to predict. **d** Co-crystal structure of DNA-bound Exd-Ubx heterodimer (PDB ID: 4CYC). Arrows indicate locations of insertion from Exd (cyan) and Ubx (green) arginine residues into the minor groove. **e**, **f** MGW predicted by the pentamer DNAshape and Deep DNAshape methods, for Exd-Ubx SELEX-seq data. Lines are calculated based on cutoff values of relative binding affinities. Darker colors in plot correspond to sequences with higher binding affinities. The comparison of the two panels shows that Deep DNAshape predicts the MGW of the flanking regions and the additional Exd minimum in MGW that the pentamer-based DNAshape method was unable to predict.

does not have this preference at the second 'AY' step[7] (Fig. 4d–f). Exd itself has another arginine in the N-terminal tail, located on the 5' side immediately upstream of the motif that inserts into the DNA minor groove[9]. Compared to the pentamer-based DNAshape, Deep DNAshape can predict DNA shape features while considering longer flanking regions and in the terminal regions of DNA target sites. Using Deep DNAshape, we revealed a narrower MGW at the correct locations according to co-crystal structures using the aligned 16-mer SELEX-seq data (Fig. 4b, c, e, f). Furthermore, for Scr, hypotheses can be made that the first 'AY' dip in MGW is required for binding, which is seen across high- to low-binding-affinity sites. The binding affinity is strengthened by a second 'AY' dip and by an additional dip for Exd binding for its N-terminus. For Ubx, unlike the pentamer-based DNAshape method, Deep DNAshape predicted the first 'AY' dip in the MGW at both A and Y bases (Fig. 4e, f). This observation is well-aligned with other Hox protein research on the influence of DNA shape[6].

### Deep DNAshape exhibits improved prediction accuracy for TF-DNA binding specificity

The mechanisms of TF-DNA readout are complex, leading to substantial research on the use of machine learning to improve the prediction accuracy for TF-DNA binding specificity[28,49,50]. In previous studies, we have successfully improved the performance of multiple linear regression tasks on TF-DNA binding affinity data by incorporating DNA shape features[27,28] (Fig. 5a). This DNA shape encoding approach reduced the degrees of freedom when using $k$-mer encoding, while maintaining the same level of performance. These models utilized both DNA sequence and shape features to achieve improved performance.

With the introduction of Deep DNAshape, which considers longer-range effects, we can now replace the DNA shape features in these machine learning models with the ones predicted by Deep DNAshape. Models using DNA shape features predicted by Deep DNAshape outperform models using features predicted with the pentamer-based DNAshape version (Fig. 5b, c). Furthermore, by using Deep DNAshape derived features in combination with fluctuation values, we were able to surpass the performance of the 3-mer model with fewer degrees of freedom (Fig. 5d). Additionally, the updated fluctuation values greatly improved the performance of the model compared to the previous SD values[28] (Supplementary Fig. 17). From a machine learning perspective, the DNA shape and fluctuation values predicted by Deep DNAshape contain more relevant information. We subsequently compared Deep DNAshape with its variants trained by different underlying data sources, finding that their performances were relatively similar (Supplementary Fig. 18). Deep DNAshape retains its performance in deeper layers that consider longer flanking regions, while the variants peak at layer 2 or 3 (See Supplementary Text).

### Deep DNAshape reveals a more conserved relationship of DNA shape in transcription start sites between *Drosophila* species

One key advantage of the DNAshape method is its ability to perform high-throughput predictions that can be easily applied to genomic-level data[51]. To assess the performance of Deep DNAshape on a large dataset, we used a dataset of TSSs from four *Drosophila* species (*D. melanogaster*, *D. simulans*, *D. sechellia*, and *D. pseudoobscura*)[52]. Previous research highlighted the importance of structural features in protein binding at TSSs[53]. Our analysis of DNA shape features at transcription start sites (TSSs) for these four fly species revealed conserved relationships in DNA shape features among these genomic regions

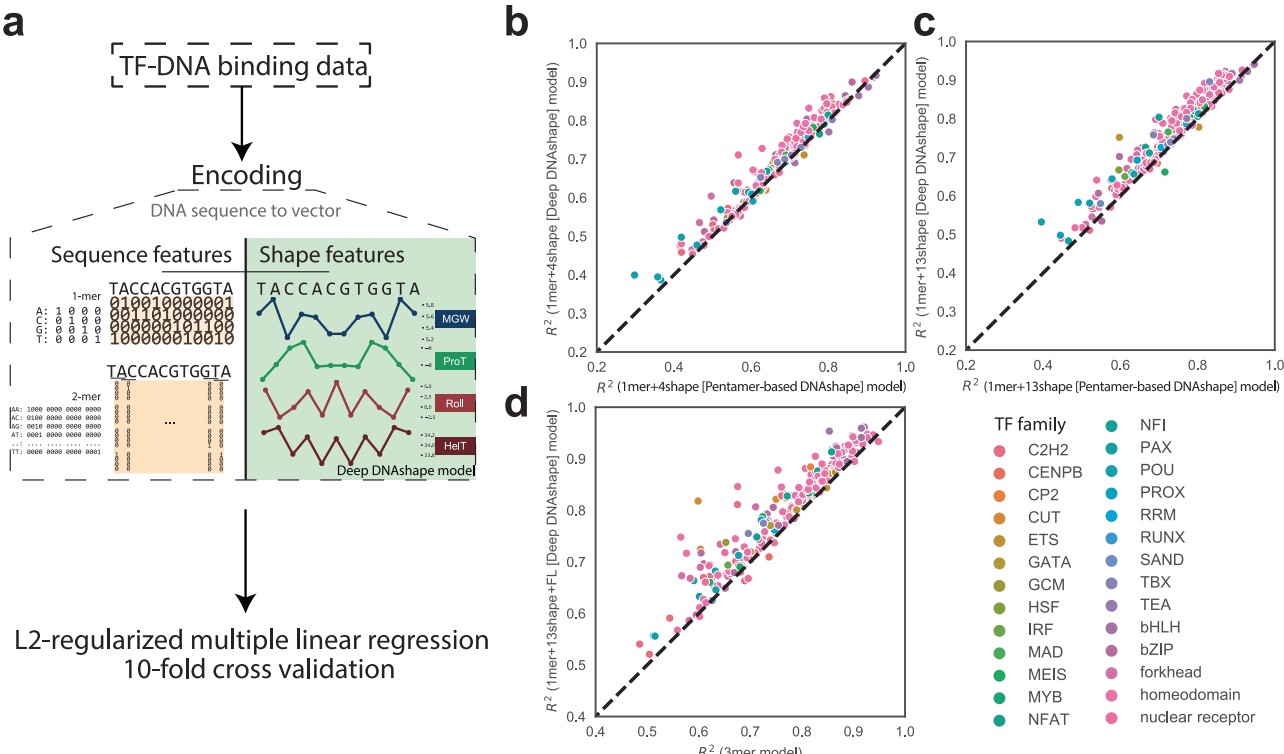

**Fig. 5 | Workflow of utilizing Deep DNAshape in machine learning modeling and performance comparison of binding specificity predictions. a** Schematic representation of predicting TF-DNA binding specificity with L2-regularized multiple linear regression model, using encoded sequence features and DNA shape features. **b–d** In vitro TF-DNA binding experimental data used in these models include gcPBM, SELEX-seq, and HT-SELEX data. **b** $R^2$ performance comparison of 1mer+4shape models between the pentamer-based DNAshape and Deep DNAshape methods. **c** $R^2$ performance comparison of 1mer+13shape models between the pentamer-based DNAshape and Deep DNAshape methods. **d** $R^2$ performance comparison between 1mer+13shape+FL models (representing the Deep DNAshape approach with all available features) and 3mer models without using DNA shape features.

despite sequence differences[51]. Data for this analysis were derived from gene expression experiments[52], and Deep DNAshape was able to process these data in a matter of minutes on a machine equipped with an NVIDIA A100. Our results showed more evolutionarily conserved relationships in the genomic structural features across these four fly species, as evidenced by the MGW, ProT, Roll, and HelT values (Supplementary Fig. 19, Supplementary Table 4).

## Discussion

In this study, we introduce Deep DNAshape, a high-throughput deep-lerarning method that accurately predicts 3D DNA structural parameters, or DNA shape, for any DNA sequence. Our pentamer-based high-throughput prediction method, DNAshape, successfully predicted DNA shape features, including MGW, inter-bp features, and intra-bp features[28,39], and was useful for studying structural TF-DNA binding mechanisms without the need of extensive molecular simulations or structural biology experiments. However, it had several major drawbacks and limitations.

First, DNAshape relied entirely on a pentamer query table to predict DNA shape features at central positions of a pentamer. Thus, to predict the DNA shape features for the pentamer core, only two bp of adjacent nucleotides in each direction were considered. The query table could not cover the full two bp of possible flanking combinations for inter-bp features, and any sequences in the flanks beyond two bp were not included in predictions. This query table may be sufficient if flanking regions over two bp away do not contribute to DNA shape at the core of the target sequence at first approximation. However, evidence has shown[48] that such extended flanking regions may affect the binding affinity and DNA shape at the core of the DNA fragment of interest. Such effects are evident in the training curve (Supplementary Figs. 2, 3) of Deep DNAshape.

Second, sequence combinations of the flanking regions outside the pentamers were not evenly distributed in the simulation data, which introduced biases towards each pentamer calculation (Supplementary Fig. 9). For example, if a pentamer is 'ACGTA' and the only heptamer in the simulation data is 'CACGTAG', assuming the third bp away from the center is still affecting the DNA shape in the core, the prediction of DNA shape features for 'ACGTA' is always biased towards 'C' and 'G' flanking the pentamer.

Our proposed Deep DNAshape method (Fig. 1) addresses the limitations of our previous method and significantly improves the ability to predict DNA shape. Unlike the pentamer-based DNAshape method that relied on pentamer query tables and sliding-window algorithms to assign DNA shape values, Deep DNAshape eliminates the use of these tools. All $k$-mers needed to occur at least once in the simulation data to complete the pentamer query table, but table completion was not sufficient to prevent statistical bias or artifacts. Even with pentamers that occurred more than 250 times[28] in the simulation data, the data still did not fully cover all hexamers (Supplementary Figs. 6 and 7), let alone heptamers (Fig. 2a, b, Supplementary Fig. 5), octamers, or longer $k$-mers. Different flanking regions have different effects on the DNA shape features at the core of DNA target sites, and these influences may be relevant to sequence components and are inferable. Deep DNAshape infers the missing values with the learnt influence of flanking regions on the core, while not harming the prediction considering short flanking regions (Supplementary Table 2).

Because the method learns the effects of flanking regions and predicts DNA shape features in a layer-by-layer manner, it is self-constrained. Each new layer infers the DNA shape features based on the DNA shape features inferred by its previous layer, while considering one more bp of flanking regions. Outputs of all layers in the training are used in calculating the loss function, enabling the model to predict DNA shape features considering any length of flanking regions, with near-zero overfitting in deeper layers (Supplementary

Figs. 2 and 3). The Deep DNAshape method establishes a general framework for predicting response variables at the single-nucleotide level, considering local environments for variable lengths of DNA, and provides valuable improvements upon the widely used DNAshape method.

DNA shape information is valuable in understanding TF-DNA binding events[1,2,28,48,54–56]. One crucial application of the Deep DNA-shape method is to investigate DNA shape preferences in TF-DNA binding. With the ability to predict accurate DNA shape features considering long-range flanking regions, our results provide insight into TF-DNA binding interactions that were previously challenging to investigate (Figs. 3 and 4). To quantify the extent to which Deep DNAshape provides more information content than our pentamer-based DNAshape method, we applied it to the prediction of TF-DNA binding specificity and found a significant improvement compared to pentamer-based DNAshape (Fig. 5). The Deep DNAshape method still operates in a high-throughput manner, allowing for the investigation of data on a genomic scale. In a brief application, we demonstrated closer evolutionary relationships in DNA shape parameters between four *Drosophila* species in TSS regions (Supplementary Fig. 19). Other recent studies[57–62] incorporating DNA shape features derived from the pentamer model will very likey benefit from using Deep DNAshape. Thus, the Deep DNAshape method unlocks a whole new level of possibilities for genomic studies of DNA shape.

Besides training the Deep DNAshape model using MC simulations as the underlying data, the model is capable of being re-trained using alternative data sources. Our benchmark analysis revealed that the Deep DNAshape variants Expt (trained using experimentally solved structural data) and MD (trained by MD simulation data) provided similar DNA shape predictions and performance metrics across multiple applications. However, the underlying data of these variants were affected by noise, artifacts, or low coverage (see Supplementary Text), making MC simulations the current optimal choice for studying effects of longer flanking regions. Future advancements in MD simulations could allow Deep DNAshape to be easily transitioned to using such data as the underlying source.

In addition to predicting DNA shape, Deep DNAshape is also a general framework for processing variable DNA lengths in a layer-by-layer manner, by using expanded neighboring-bp information in DNA sequences and by inferring response variables (e.g., nucleosome positioning or 3D genome interaction) at the single-nucleotide level. Other research that utilizes DNA shape as a feature would likely benefit from the use of the Deep DNAshape method. To further improve Deep DNAshape, one could focus on generating additional simulation data, optimizing the DNA shape inference equation or network architecture, and expanding the model to include chemically modified bp[63] or mismatched bp[11], if such data could be acquired on a large scale.

## Methods

### DNA structural simulations and DNA shape analyses

DNA sequences with variable lengths are initialized and simulated by a Monte-Carlo (MC) method[27]. After filtering out artifacts, the total number of valid DNA simulations is 2,121[27]. DNA shape features are then calculated by Curves (version 5.3)[12,25] and minor groove width (MGW) is symmetrized with respect to each bp[27]. Data are pooled into a training file that includes all DNA sequences and their corresponding DNA shape values. DNA shape fluctuation values are also calculated by Curves through analyzing the sampling process from the MC simulations. Fluctuation values represent the variance that occurred during simulations. These values can be viewed as a correlated measure of DNA conformational flexibility of an individual DNA molecule. In this study, the addition of '-FL' to a DNA shape feature indicates the fluctuations (FL) of that specific DNA shape feature. Note that DNA shape can be derived from other data sources (see Supplementary Methods).

## Definitions of DNA shape features

DNA shape features are used to describe DNA structure numerically, in a bp-to-bp manner. The set of DNA shape features used includes six inter-bp features, six intra-bp features, and two minor groove features. Additional DNA shape features include helical axis features and backbone torsion features, which can be analyzed and predicted if supplied to the model. Inter-bp features are used to describe translational distances (in Å) and rotational angles (in °) between adjacent bp. Specifically, the six inter-bp features are 'Shift', 'Slide', 'Rise', 'Tilt', 'Roll', and 'Helix twist (HelT)' (Fig. 1g). The intra-bp features describe the translational distances and rotational angles between two bases in a single bp. The six intra-bp features are 'Shear', 'Stretch', 'Stagger', 'Buckle', 'Propeller twist (ProT)', and 'Opening' (Fig. 1g). The minor groove features describe the groove geometry and electrostatic potential in the center of the minor groove. The groove features used in this study are 'Minor groove width (MGW)' (Fig. 1g) and 'Electrostatic Potential (EP)'[29]. For detailed information, refer to[28].

## Pre- and post-processing of DNA shape values

To ensure robustness in our DNA shape analysis, we utilized a normalization method to account for different ranges of minimum and maximum values of DNA shape features. We compensated for extreme values (e.g., from simulation artifacts) by using the following normalization equation:

$$\hat{S} = (S - \widetilde{S})/(S_{1st} - S_{99th}) \tag{1}$$

Here, $S$ represents the DNA shape feature analyzed from the MC simulation. $\widetilde{S}$ denotes the median of the DNA shape feature values within the dataset, while $S_{1st}$ and $S_{99th}$ mark the first percentile and last percentile of the sorted DNA shape feature values, respectively. After normalization, $\hat{S}$ may still exceed the range of −1 to +1 for extreme values. However, during the training process, the tanh activation function enforces a strict upper and lower bound of $S_{1st}$ and $S_{99th}$, ensuring that the output remains within a normalized range of −1 to +1. In postprocessing of the model output, given the normalized DNA shape values predicted by our model, $S$ can be directly computed from $\hat{S}$ using the reverse of the above equation given knowledge of $\widetilde{S}$, $S_{1st}$, and $S_{99th}$.

## Definition of the DNA shape layer

The DNA shape layer is used to treat linear DNA sequences as double-linked nodes or a graph, where each node can be a single nucleotide or dinucleotide. Each node is connected by its 5' node and 3' node through a forward and a backward 'bond' (edge). To compute the output features $\widetilde{X}_i$ of the shape layer, features are gathered for each node from the previous node ($X_{i-1}$), current node ($X_i$), and next node ($X_{i+1}$). $\widetilde{X}_i$ is computed through the following equation,

$$\widetilde{\boldsymbol{X}}_i = f\left(\sigma(\boldsymbol{\lambda} \cdot \boldsymbol{X}_i + \boldsymbol{\alpha}), \boldsymbol{X}_i\right) \tag{2}$$

where,

$$\boldsymbol{\lambda} = \boldsymbol{\omega}_1 \cdot \boldsymbol{X}_{i-1} + \boldsymbol{\omega}_2 \cdot \boldsymbol{X}_{i+1} + \boldsymbol{B}_1$$

$$\boldsymbol{\alpha} = \boldsymbol{\theta}_1 \cdot \boldsymbol{X}_{i-1} + \boldsymbol{\theta}_2 \cdot \boldsymbol{X}_{i+1} + \boldsymbol{B}_2 \tag{3}$$

$f$ is a trainable gated recurrent unit (GRU) cell with a sigmoid recurrent activation function and tanh activation function. $\sigma$ is ReLu activation followed by batch normalization function. $\omega_1$, $\omega_2$, $\theta_1$, $\theta_2$, $B_1$ and $B_2$ are trainable variables to be learnt from the dataset by the model. The self-shape layer is a one-dimensional convolutional layer that transforms the feature number to match later DNA shape layers. The feature dimension remains the same for all DNA shape layers.

## Dropout layers and average layers

For each DNA shape layer, a feature vector is generated for every node in the DNA sequence. To prevent overfitting, the feature vector passes through a dropout layer. During prediction, the dropout layer is not used. Finally, the feature vectors are averaged into a single value and post-processed to remove the effects of normalization before making a prediction.

## DNA sequence encoding

To represent the DNA sequences, four characters, 'A', 'C', 'G' and 'T' are used, and are one-hot encoded into four arrays as [1,0,0,0], [0,1,0,0], [0,0,1,0] and [0,0,0,1]. For example, a sequence, 'ACGTGCG', is represented as

$$\begin{bmatrix} A \\ C \\ G \\ T \\ G \\ C \\ G \end{bmatrix} = \begin{bmatrix} 1 & 0 & 0 & 0 \\ 0 & 1 & 0 & 0 \\ 0 & 0 & 1 & 0 \\ 0 & 0 & 0 & 1 \\ 0 & 0 & 1 & 0 \\ 0 & 1 & 0 & 0 \\ 0 & 0 & 1 & 0 \end{bmatrix} \tag{4}$$

To represent the DNA sequences as dinucleotides, one-hot encoding for dinucleotides was used. Each dinucleotide will be assigned a vector with 16 binary values to represent 16 possible dinucleotide combinations. The same example 'ACGTGCG' will be encoded as

$$\begin{bmatrix} AC \\ CG \\ GT \\ TG \\ GC \\ CG \end{bmatrix} = \begin{bmatrix} 0100,0000,0000,0000 \\ 0000,0010,0000,0000 \\ 0000,0000,0001,0000 \\ 0000,0000,0000,0010 \\ 0000,0000,0100,0000 \\ 0000,0010,0000,0000 \end{bmatrix} \tag{5}$$

An unknown nucleotide 'N' can also be represented this way, where 'N' will be a vector by taking the average values upon all possible values. In the implementation of Deep DNAshape, to balance the degree of freedom of the terminal bases, 'N' caps are added to both terminals, but they are removed in the final prediction. These 'N' caps have a 3' bond and a 5' bond connected to themselves.

## Deep DNAshape model design and learning objectives

The model is designed for step-by-step, expandable learning of DNA shape features for any given DNA sequence, with input as one-hot-encoded DNA sequences and output as predicted DNA shape features for each position on the sequence. Inter-bp DNA shape features are encoded as dinucleotides, and sequences are represented as linear double-linked nodes. Mean absolute error (MAE) is used as the loss function, calculated between predicted and MC-simulated DNA shape values from output layers. A postprocessing step is used to recover DNA shape values from normalization. Individual models are trained for each DNA shape feature, with one 'self' convolutional layer for input and seven 'shape layers' following it.

## Hyperparameter search

Hyperparameters for the Deep DNAshape model include learning rate, optimizer, filter size, and others. These hyperparameters are grid searched for each DNA shape feature and evaluated on a separate training and validation dataset. The best-performing hyperparameters are selected and applied to all DNA shape features. After hyperparameter searches, the model parameters are set as follows: number of

shape layers: 7, learning rate: 0.05, number of epochs: 1500, optimizer: stochastic gradient descent (SGD) with momentum 0.95, dropout ratio: 0.5 and filter size: 64.

### Data of TF-DNA binding assays

We collected and used relative binding affinity data captured in multiple experiments, the same as were used in[28]. The genomic-context PBM data include human TFs c-Myc, Max, and Mad2[64], with 'Max' representing a Max-Max homodimer, 'c-Myc' a c-Myc-Max heterodimer, and 'Mad2' a Mad2-Max heterodimer. HT-SELEX data include many TFs in multiple TF families[65]. SELEX-seq data include TFs in the Hox family from *Drosophila*[5,7].

### L2-regularized multiple linear regression model for TF-DNA binding prediction

The L2-regularized multiple linear regression model is designed to predict relative TF-DNA binding specificity from the data of the TF-DNA binding assays[28]. The model encodes DNA sequences as $k$-mer ($k = 1,2,3$) sequence features and any number of DNA shape features. Unless otherwise specified, '4shape' indicates four shape features including MGW, ProT, Roll, and HelT, and '13shape' indicates MGW plus the 6 inter-bp features and 6 intra-bp features. The DNA shape features are normalized according to the minimum, maximum, and standard deviation seen in the dataset. Input data are separated into 10 folds. In each fold of the training and test data, another 10-fold cross validation is used in the training data to select lambda values for the L2 term. The model then uses the selected lambda to fit the training data and predict the values for the test data. In the end, 10 folds of predictions are combined to assess the model performance as $R^2$. The number of sequence features, for sequences with length $n$, is $4n$ for 1-mers, $16 \cdot (n-1)$ for 2-mers, and $64 \cdot (n-2)$ for 3-mers. The number of shape features, for sequences with length $n$, is $n$ for each bp feature and groove feature, and $n-1$ for each bp step feature.

### Reporting summary

Further information on research design is available in the Nature Portfolio Reporting Summary linked to this article.

## Data availability

TF-DNA binding datasets were derived from public resources (see Methods, Data of transcription factor (TF)-DNA binding assays). Raw data for the underlying training datasets (MC, MD and Expt) were sourced from reference[27] and public databases (see Supplementary Methods for details). PDB IDs used in the main text are 1AN2, 2R5Z and 4CYC. Process data of TF-DNA binding and underlying training data are deposited at Zenodo <https://doi.org/10.5281/zenodo.10403307>. Source data are provided with this paper.

## Code availability

All code was implemented in Python. All code related to training, prediction, and the pre-trained models – as well as an executable package for predicting DNA shape features from any DNA sequence – can be found at https://github.com/JinsenLi/deepDNAshape[66]. All source code is provided under an BSD-3-Clause software license.

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

## Acknowledgements

The authors acknowledge Tianyin Zhou for the development and validation of the pentamer-based DNAshape method. This work was supported by the National Institutes of Health [grant R35GM130376 to R.R.] and the Human Frontier Science Program [grant RGP0021/2018 to R.R.].

## Author contributions

J.L., T.C. and R.R. conceived the project. J.L. and T.C. designed the model architecture. J.L. performed all training, evaluations, and analyses. J.L. wrote the manuscript with help from T.C. and R.R. The project was supervised by R.R.

## Competing interests

The authors declare no competing interests.

## Additional information

**Peer review information** : *Nature Communications* thanks Maria Poptsova and the other, anonymous, reviewer(s) for their contribution to the peer review of this work. A peer review file is available.

