## [Peer review file · Nature Communications]

REVIEWER COMMENTS

Reviewer #1 (Remarks to the Author):

In the presented study "Deep DNASHape: a deep learning method to predict DNA structure considering the influence of extended flanking regions" the authors present a deep learning model with specialized architecture to predict DNA shape parameters from DNA sequence. The work seems to be a logical continuation of methods developed earlier by the same group and the presented Deep DNA Shape method is compared with the authors' DNA Shape method. DNA Shape method uses precomputed query pentamer table containing shape features for all possible pentamers. The shape features (minor groove width and 12 others such as Tilt, Shift, Slide etc., 13 in total) were computed with Curves program from sequences generated randomly (Monte Carlo (MC) approach) so that each pentamer was present around 40 times or so in the generated set. The drawback of this pentamer approach, as authors refer it in the paper, is that it is restricted to the sequence length of 5 – only flanks of 2 bp length are considered. To apply the same approach to +/-3bp flanks (septamers) would be computationally difficult (though the present day computational power should handle it). MC approach also allowed for calculating variance or standard deviation of the features that can be attributed to flexibility of DNA and it is a useful property to consider.

The goal of the presented model any DNA shape feature (including fluctuations) for any length of DNA sequence and considering more flanking bases as compared to the implemented DNA Shape.

For that the authors developed a specialized network architecture with the goal to predict any DNA shape feature for any length of DNA sequence. It inputs one-hot-encoded DNA sequence (can be substituted with di-nucleotide one-hot-encoded matrix) and outputs features for each position in a sequence. After the first 1D convolution layer there follows a set of so-called Shape layers. Each shape layer consists of GRU-cells with a cell being one nucleotide. Each consequent Shape layer adds 1bp to flanks (and here I could not understand how it was implemented - see the comments below). Each feature is predicted individually with individually trained model.

The predictions generated by Deep DNA Shape were compared to DNA Shape on different TFs such as Max, Hox-TF.

Overall I think the study is very interesting and useful. It adds on previous studies. And also it explores the power of Deep learning model to predict shape parameters from sequence. The other useful outcome that the model predicts fluctuations for every feature. The usage of the model is demonstrated.

Major comments

1. The formula the author provided for DNA shape layer takes into account only previous and next node (=nucleotide). The shape layer consists of GRU cells. It inputs a sequence and outputs feature. The next shape layer takes as input sequence and the features generated on layer earlier. And here I do not understand how larger than 1 bp flanks are taken into account if the formula includes only one previous and one next nucleotide. Does it mean that for 2 bp influence the formula will include $i-2$, $i-1$, $i+1$, $i+2$ nucleotide? Is it a play with filter lengths? – Additional explanation is required.

2. Fig 1c. I read the arrows between shape layers as the output of one layer is used for in input of the next layer. However the input is sequence and the output is features. I understood it as that the features generated on one layer are passed to the next layer as the feature of the sequence. If so then the schema does not tell it. There should be a connection from generated features from the first Shape layer passing to the second Shape layers, etc. Or I did not understand the schema correctly. Fig 1c. Legend. "features for each individual position are passed to its nearby two positions" – when and how more bp in flanks are taking into account?

3. The authors always demonstrate only MGW feature. Are others have similar predictive value? Supplementary Fig 2-4 show the other three feature but for comparison of the methods not for prediction of binding sites. It would be useful if authors demonstrate cases when TF binding well correlates with this other than MGW plot. Otherwise it is not clear why one should bother to predict other 12 features.

4. Example Fig 3d - I do not see difference between pentamer DNA shape and Deep DNA Shape.

5. Did authors try just CNN or LSTM or hybrid CNN+LSTM that are actively used today. I am curious why the choice was GRU?

Minor

I could not find the program Curves specifically "5.3". – the references are for 1988 paper. Did authors mean Curves+?

No references to DeepDNABind

Reviewer #2 (Remarks to the Author):

Rohs and coworkers present an interesting work intended to show how flanking regions can influence the binding of transcription factors in particular those whose binding is dominated by shape effects. The idea is appealing and indeed, many transcription factors have long recognition sequences and even those that bind in the pentamer level can feel the neighboring effect.

The authors move their DNAshape protocol, presented already many times coupled with different shape descriptors to consider longer effects that are obtained by ML-training of shape descriptors, which in principle, provided the set of simulations used for training is large enough, can be extended to any reasonable length of the duplex (they cut however to the 7th base pair).

The paper is nice and the approach is interesting, but I do not think it represents a dramatic advantage with respect to previous development from the group and other ML approaches developed by others.

The benchmark of the method is rather limited, only one type of experimental data is considered, and no comparison is made with all alternative approaches to predict TF binding in vitro and in vivo.

The source of structures to derive descriptors is the same than in the original SHAPE paper, very simple Monte Carlo simulations using a very simplified implicit model system and restricted polymers with closure conditions. This level of quality is below what is considered today the state of the art in simulation. No systematic evaluation of the quality of the MC structure, and accordingly of the data that is used for training.

REVIEWER COMMENTS

Reviewer #1 (Remarks to the Author):

In the presented study “Deep DNashape: a deep learning method to predict DNA structure considering the influence of extended flanking regions” the authors present a deep learning model with specialized architecture to predict DNA shape parameters from DNA sequence. The work seems to be a logical continuation of methods developed earlier by the same group and the presented Deep DNA Shape method is compared with the authors' DNA Shape method. DNA Shape method uses precomputed query pentamer table containing shape features for all possible pentamers. The shape features (minor groove width and 12 others such as Tilt, Shift, Slide etc., 13 in total) were computed with Curves program from sequences generated randomly (Monte Carlo (MC) approach) so that each pentamer was present around 40 times or so in the generated set. The drawback of this pentamer approach, as authors refer it in the paper, is that it is restricted to the sequence length of 5 – only flanks of 2 bp length are considered. To apply the same approach to +/-3bp flanks (septamers) would be computationally difficult (though the present day computational power should handle it). MC approach also allowed for calculating variance or standard deviation of the features that can be attributed to flexibility of DNA and it is a useful property to consider.

AUTHORS: We thank this Reviewer for considering this work as logical continuation of our previous work on DNA shape prediction and summarizing how the new method, Deep DNashape, overcomes limitations of our previous pentamer-based methods.

The goal of the presented model any DNA shape feature (including fluctuations) for any length of DNA sequence and considering more flanking bases as compared to the implemented DNA Shape.

For that the authors developed a specialized network architecture with the goal to predict any DNA shape feature for any length of DNA sequence. It inputs one-hot-encoded DNA sequence (can be substituted with di-nucleotide one-hot-encoded matrix) and outputs features for each position in a sequence. After the first 1D convolution layer there follows a set of so-called Shape layers. Each shape layer consists of GRU-cells with a cell being one nucleotide. Each consequent Shape layer adds 1bp to flanks (and here I could not understand how it was implemented - see the comments below). Each feature is predicted individually with individually trained model.

The predictions generated by Deep DNA Shape were compared to DNA Shape on different TFs such as Max, Hox-TF.

AUTHORS: Yes. Thanks for this accurate description of the method and its advances over existing approaches. It is important to emphasize that Deep DNashape is no longer limited by the underlying pentamer of the previous DNashape and can predict DNashape features for DNA of any length including flanking regions. The methodological implementation was clarified in the revised manuscript.

Overall I think the study is very interesting and useful. It adds on previous studies. And also it explores the power of Deep learning model to predict shape parameters from sequence. The other useful outcome that the model predicts fluctuations for every feature. The usage of the model is demonstrated.

AUTHORS: We thank the Reviewer for finding this study interesting and useful. We demonstrated that the deep learning model is applicable to DNA shape prediction and we also applied predictions of the current model to multiple biological questions, including transcription factor binding (e.g., Hox, Max) and genome wide shape analysis (transcription start sites).

Major comments

1. The formula the author provided for DNA shape layer takes into account only previous and next node (=nucleotide). The shape layer consists of of GRU cells. It inputs a sequence and outputs feature. The next shape layer takes as input sequence and the features generated on layer earlier. And here I do not understand how larger than 1 bp flanks are taken into account if the formula includes only one previous and one next nucleotide. Does it mean that for 2 bp influence the formula will include $i-2$, $i-1$, $i+1$, $i+2$ nucleotide? Is it a play with filter lengths? – Additional explanation is required.

AUTHORS: We apologize that we were not clear enough in describing the architecture of the deep learning approach in our original submission. We have now added a detailed description and additional supplementary figure, Fig. S1, that describes the shape layers and how they are used in the method. Indeed, as the Reviewer has deduced, the shape layer does primarily focus on the immediately preceding and succeeding nodes. However, our message-passing architecture allows for the accumulation of features calculated at each node, which inherently includes information from adjacent nodes. This cumulative knowledge effectively enables the model to account for an additional base pair in the flanking regions during each subsequent layer calculation. This occurs as we gather information from previous and subsequent nodes. However, it is also important to note that due to the gating mechanism inherent in GRU cells, this influence from neighboring nodes should progressively diminish with distance. This design aligns with our underlying assumption about the influence of extended flanking regions on the core DNA shape. We hope this additional explanation, alongside the new supplementary Fig. S1, addresses the Reviewer's query.

2. Fig 1c. I read the arrows between shape layers as the output of one layer is used for in input of the next layer. However the input is sequence and the output is features. I understood it as t that the features generated on one layer are passed to the next layer as the feature of the sequence. If so then the schema does not tell it. There should be a connection from generated features from the first Shape layer passing to the second Shpae layers, etc. Or I did not understand the schema correctly.

Fig 1c. Legend. “features for each individual position are passed to its nearby two positions” – when and how more bp in flanks are taking into account?

AUTHORS: We appreciate the Reviewer's feedback and have revised Fig. 1 to address this point, in addition to including a new supplementary Fig. S1 that describes our model's architecture in more detail. The modified figure now accurately labels the layers and illustrates the data structure. As the Reviewer correctly pointed out, the output features of one shape layer serve as input to the next. These features also act as the input for generating DNA shape predictions through the dropout (only during training) and average layer situated on the side. Here, predictions are assessed, and gradients are calculated. Our new supplementary Fig. S1 and revised Fig. 1 now provide a more precise description of this process, detailing how each layer operates and how flanking regions are taken into account through our iterative message-passing system. We hope this will clarify this point and apologize for any confusion this might have caused.

3. The authors always demonstrate only MGW feature. Are others have similar predictive value? Supplementary Fig 2-4 show the other three feature but for comparison of the methods not for prediction of binding sites. It would be useful if authors demonstrate cases when TF binding well correlates with this other than MGW plot. Otherwise it is not clear why one should bother to predict other 12 features.

AUTHORS: This is a good point. The MGW is indeed a structural feature that spans multiple base pairs and is therefore more substantially influenced by neighboring base pairs. Intra-base-pair features are predominantly influenced by the chemical identity of just the base pair and inter-base-pair features are primarily influenced by the stacking geometry between just two adjacent base pairs. Although these shape features are all influenced by neighboring regions, they are less affected by multiple base pairs than MGW, as can be seen in the model training curve [Fig S2].

To provide additional evidence for the predictive value of other features, we have included another supplementary figure (Fig. S14) demonstrating the DNA shape feature 'Roll' in a different representation, to complement Fig. 3. The figure depicts the correlation between core DNA shape (affected by flanks) and binding affinity, even when the core motif remains constant. This is an insight that the previous DNashape method, which relies on a pentamer query table, cannot provide, as it would always predict the same DNA shape values, regardless of the flanking regions.

Moreover, as shown in Fig. 5, incorporating all DNA shape features into a machine learning model can yield robust results. Using DNA shape features have been shown to boost many machine learning applications [PMID: 25775564, 29165643, 31634140, 34929739, 33767912, 35015646, 34882561, etc.], which shall be boosted even further upon using Deep DNashape. In addition, given our prediction of 12 inter- and intra-base-pair features, one can use methods such as X3DNA to rebuild the 3D DNA structure, showcasing another potential application of these predictions. We hope that this point addresses the Reviewer's question about the importance of predicting other DNA shape features.

4. Example Fig 3d - I do not see difference between pentamer DNA shape and Deep DNA Shape.

AUTHORS: Thanks for this comment. Figure 3 e-h and Supplementary Figs. S14-15 were designed to highlight the differences between the pentamer DNA shape and Deep DNA Shape predictions. These differences are primarily evident in the distribution of the predicted values and the impact of the flanking regions on the predictions.

Using the previous pentamer-based DNashape model, when the dataset was filtered to only include the E-box motif CACGTG (as shown in these figures), the predicted DNA shape was always identical, making it non-distinguishable and offering no additional structural information. However, Deep DNashape resolves this issue and overcomes this limitation by considering the influence of extended flanking regions.

As an example, Deep DNashape yields new insights into the binding mode of the bHLH family, as shown in Fig. S16 (included below). These results reveal novel mechanisms of the readout of DNA by the bHLH family, which warrant further investigation. We hope this explanation clarifies the distinction and provides added value of the use of Deep DNashape.

5. Did authors try just CNN or LSTM or hybrid CNN+LSTM that are actively used today. I am curious why the choice was GRU?

AUTHORS: As suggested by the Reviewer, we have experimented with various model architectures before choosing the GRU. When designing our model, we considered the unique aspects of predicting DNA structural parameters and the several limitations associated with the training data. We contemplated how biophysical forces pass through a DNA polymer, recognizing that different base pairs would exhibit different behaviors. Our conclusion was that the interactions of neighboring base pairs should decrease with distance and that specific combinations of base pairs might function differently. Initially, we tried to run a message-passing RNN with either LSTM or GRU multiple times per shape layer to include more base pair information. However, these models did not perform well. After careful refinement of our model architecture, we opted for a single-pass GRU layer per shape layer. Without a GRU/LSTM layer, or with LSTM, the models did not perform as well as with GRU (Refer to the training curve of the MGW feature for an example. A pink dashed line illustrates the lower MAE achieved by the GRU version. The black horizontal line represents the benchmark of the pentamer DNashape).

In our selected design, the GRU cell functions more like a gate compared to the GRU layers used in recurrent neural networks. The GRU decides whether considering an additional base pair of features is advantageous and what proportion of these effects should be taken into account. We hope that this clarifies our choice of GRU.

Minor

I could not find the program Curves specifically “5.3”. – the references are for 1988 paper. Did authors mean Curves+?

AUTHORS: Curves 5.3. is at the core of the original DNashape method and it is the underlying approach for Curves+. We consider it therefore appropriate to cite as original literature. It also provides the most detailed description of the Curves algorithm and standard reference frame. This said, we accept the point made by the Reviewer that the more modern adoption of Curves 5.3. in the Curves+ algorithm should be cited, and we now include this reference to Curves+. Curves+ is more accessible through a modern webserver, although the original Curves 5.3. algorithm allowed for the calculation of additional structural parameters and choices in how helical axes are chosen and defined. As such, Curves 5.3., is the parent algorithm for both Curves+ (which was used for analyzing MD data) and DNashape (used for analyzing MC data) and it is appropriate to cite all three of these mathematically related approaches.

No references to DeepDNABind

AUTHORS: We thank the Reviewer for this suggestion. However, unfortunately, we were unable to identify a method named “DeepDNABind”. If they meant “Deepbind”, we have cited this work in our study.

Reviewer #2 (Remarks to the Author):

Rohs and coworkers present an interesting work intended to show how flanking regions can influence the binding of transcription factors in particular those whose binding is dominated by shape effects. The idea is appealing and indeed, many transcription factors have long recognition sequences and even those that bind in the pentamer level can feel the neighboring effect.

The authors move their DNAscape protocol, presented already many times coupled with different shape descriptors to consider longer effects that are obtained by ML-training of shape descriptors, which in principle, provided the set of simulations used for training is large enough, can be extended to any reasonable length of the duplex (they cut however to the 7th base pair).

AUTHORS: We thank this Reviewer for their appreciation of our DNA shape methods and related insights in transcription factor-DNA binding. This new work is indeed overcoming limitations that our previous pentamer-based method had. The new method also allows DNA shape predictions that are no longer restricted to an underlying model with a length restriction (also not limited to 7 base pairs).

The paper is nice and the approach is interesting, but I do not think it represents a dramatic advantage with respect to previous development from the group and other ML approaches developed by others. The benchmark of the method is rather limited, only one type of experimental data is considered, and no comparison is made with all alternative approaches to predict TF binding *in vitro* and *in vivo*.

AUTHORS: We respectfully disagree – In all our past efforts and other ML approaches that utilized DNA parameters based on our pentamer method, there was a constraint: the reliance on a pentamer query table. Contrary to an impression that this was a biologically informed choice, it was purely a technical limitation. It sometimes led to conclusions possibly biased by this design.

Our current approach overcomes this limitation. By moving away from the pentamer query table, we introduced an unbiased method that, for the first time, offers insights potentially closer to biological reality. For the first time, we have a method at hand, that allows us to query variations of structural parameters within an E-box (as one example) or within flanking base pairs (as another example). We are confident that any protein-DNA prediction model which integrated our previous DNAscape method stands to benefit from this novel approach.

We consciously chose *in vitro* binding data as our focus because it offers a purer form of data, allowing us to examine the biophysical interactions between DNA and proteins without additional interference of other *in vivo* cellular factors, such as nucleosome positioning, co-factors, DNA methylation, histone modification, etc. These factors can significantly affect binding specificity and affinity between protein and DNA. We consider this initial study a baseline study without cellular components but follow up studies by other researchers will certainly include *in vivo* data.

Benchmarking our model on a different dataset is a valid suggestion. We had initially contemplated including a dataset from experimentally resolved structures, but such a dataset could be heterogeneous and contain irregularities and artifacts, necessitating curation effort. Therefore, we chose to include only MC data as the underlying dataset of our primary Deep DNAscape method in the first submission. However, based on the Reviewer's suggestion, we also attempted learning DNA shape features acquired from experimentally (Expt) solved structures and Molecular Dynamics (MD) simulations for additional variants of our method, Deep DNAscape (Expt) and Deep DNAscape (MD). We have added sections in the supplementary material and related parts in the main manuscript discussing this. In general, Deep DNAscape can learn DNA shapes acquired from experimental data (Fig. S4). These experimental data were lightly filtered (see description in Supplementary Information) to remove deformed and overrepresented structures. Although it is clearly not ideal to compare DNA shape features of free DNA

fragments in solvent with the DNA structures from protein-bound and protein-deformed DNA molecules in a co-crystal, the majority of DNA shape features predicted by Deep DNASHape (Experimental) still correlate well with those predicted by Deep DNASHape (MC) (Table S3) and perform similarly in machine learning applications (Fig. S18).

The source of structures to derive descriptors is the same than in the original SHAPE paper, very simple Monte Carlo simulations using a very simplified implicit model system and restricted polymers with closure conditions. This level of quality is below what is considered today the state of the art in simulation. No systematic evaluation of the quality of the MC structure, and accordingly of the data that is used for training.

AUTHORS: We appreciate the Reviewer's point of view on MC simulations. We agree that MC simulations, at the surface, use a simplified approach compared to MD simulations – MC simulations calculate energy with an implicit solvent model versus forces in explicit water necessary for MD simulations. However, this methodological distinction doesn't necessarily position one as superior to the other.

MC methods do not require force calculations, allowing for the efficient use of an implicit solvent model. This can be advantageous, as it gives modern force fields the opportunity to sample more efficiently intra-molecular interactions such as base stacking and not be dominated by solvent-solute interactions. For instance, MC simulations with rather earlier versions of the AMBER force field could reproduce the Helical Twist of DNA more accurately than MD simulations (Rohs et al., Structure 2005).

Also, very importantly, in nucleic acids, backbone torsion angles such as (α , γ) pairs undergo flips that can only be easily reversibly sampled in MC simulations. Here, the polymer with closure conditions is key as all chain closure conditions can be energetically considered in an MC algorithm, whereas an MD simulation is often unable to overcome a torsion angle energy barrier such as the (α , γ) flip in the phosphodiester backbone. Therefore, the MC approach has an advantage in sampling the very flexible nucleic acid structure and all its conformational variants. Doing this in implicit solvent makes the sampling of large conformational transitions even more efficient as water molecules will not hinder the movements as they would in an explicit solvent environment. This said, future work and large-scale MD simulation projects might change this perspective, but I do not see the field there yet.

As for validation, we agree with the Reviewer. The more validation, the better. The MC-based DNASHape method underwent rigorous validation through experimental data from X-ray crystallography, NMR spectroscopy, and hydroxyl radical cleavage measurements (Zhou et al. NAR 2013), and compared with X-ray crystallography and Molecular Dynamics data, including machine learning approaches (Li et al. NAR 2017).

Obtaining DNA shape data for a significant amount of DNA sequences using MD simulations remains challenging. We did attempt to leverage MD simulations by acquiring as much data as possible from the Parmbsc1 database and trained our Deep DNASHape model with this, compared to MC, sparse data. The results were relatively good (Fig. S4 and Table S3); however, the model did not seem to learn much additional information from longer neighboring effects due to the low coverage of MD data (Fig. S18).

This in turn reinforces the advantage of the Deep DNASHape architecture and justifies our use of MC data as the underlying training data.

Acquiring a large dataset of DNA structures via MD simulations would require large collaborations. To the best of our knowledge, the ABC consortium is already conducting many more MD simulations to cover all hexamers. We appreciate their effort and look forward to their results. We believe that once these results are available, Deep DNASHape will be capable of learning from this data and provide a version of Deep DNASHape with DNA shape data acquired from MD simulations as the underlying training data.

This said, the actual data used can and likely will be updated in future Deep DNASHape models. This paper, however, introduces an approach to deduce DNA structural features of extended fragments without the limitation of query table of any length, thus overcoming a substantive limitation of previous approaches.

We thank both Reviewers for their important contributions and suggestions, which have substantially improved our work and manuscript.

REVIEWERS' COMMENTS

Reviewer #1 (Remarks to the Author):

The authors addressed all the major issues and considerably revised the manuscript. In the current form I recommend it for publication.

Reviewer #3 (Remarks to the Author):

I have read the manuscript carefully and have gone over the reviewer comments and the author response. In my opinion every one of the issues raised by the reviewers have been adequately addressed. I was particularly taken by the response regarding MD v.s. MC simulations. It is clear that despite the simpler model, MC simulations have successfully made discoveries about the role of DNA shape that are still not addressable with MD simulations. The authors make this point very clearly.

Authors' Response to Reviewers' comments

Second Revision of Manuscript **NCOMMS-23-17023A**

REVIEWERS' COMMENTS

Reviewer #1 (Remarks to the Author):

The authors addressed all the major issues and considerably revised the manuscript. In the current form I recommend it for publication.

Reviewer #3 (Remarks to the Author):

I have read the manuscript carefully and have gone over the reviewer comments and the author response. In my opinion every one of the issues raised by the reviewers have been adequately addressed. I was particularly taken by the response regarding MD v.s. MC simulations. It is clear that despite the simpler model, MC simulations have successfully made discoveries about the role of DNA shape that are still not addressable with MD simulations. The authors make this point very clearly.

AUTHORS: We thank both Reviewers for their important contributions and suggestions, which have substantially improved our work and manuscript.